behaviour, evolution, neuroscience

pair bond, sex differences, species differences, butterflyfish, nonapeptides

**Author for correspondence:**
Lauren A. O'Connell
e-mail: loconnel@stanford.edu

# Gene expression correlates of social evolution in coral reef butterflyfishes

Jessica P. Nowicki[1,2], Morgan S. Pratchett[1], Stefan P. W. Walker[1], Darren J. Coker[1,3] and Lauren A. O'Connell[2]

[1]ARC Centre of Excellence for Coral Reef Studies, James Cook University, Townsville, Queensland 4810, Australia
[2]Department of Biology, Stanford University, 371 Jane Stanford Way, Stanford, CA 94305, USA
[3]Red Sea Research Center, Division of Biological and Environmental Science and Engineering, King Abdullah University of Science and Technology, Thuwal 23955-6900, Saudi Arabia

JPN, 0000-0002-6373-8761; LAO, 0000-0002-2706-4077

Animals display remarkable variation in social behaviour. However, outside of rodents, little is known about the neural mechanisms of social variation, and whether they are shared across species and sexes, limiting our understanding of how sociality evolves. Using coral reef butterflyfishes, we examined gene expression correlates of social variation (i.e. pair bonding versus solitary living) within and between species and sexes. In several brain regions, we quantified gene expression of receptors important for social variation in mammals: oxytocin (*OTR*), arginine vasopressin (*V1aR*), dopamine (*D1R*, *D2R*) and mu-opioid (*MOR*). We found that social variation across individuals of the oval butterflyfish, *Chaetodon lunulatus*, is linked to differences in *OTR*, *V1aR*, *D1R*, *D2R* and *MOR* gene expression within several forebrain regions in a sexually dimorphic manner. However, this contrasted with social variation among six species representing a single evolutionary transition from pair-bonded to solitary living. Here, *OTR* expression within the supracommissural part of the ventral telencephalon was higher in pair-bonded than solitary species, specifically in males. These results contribute to the emerging idea that nonapeptide, dopamine and opioid signalling is a central theme to the evolution of sociality across individuals, although the precise mechanism may be flexible across sexes and species.

## 1. Introduction

Animals display spectacular diversity in sociality (i.e. affiliative social behaviour [1]), prompting fundamental questions about how it arises and evolves. A striking example that has garnered considerable research attention is pair-bonded versus solitary living (herein referred to as 'social variation' or 'variation in sociality'), which varies on an individual and species level in several vertebrate lineages [2–6]. Understanding the mechanisms of individual social variation within a species's population is important, as this is the level on which natural selection acts to drive evolutionary change. Equally important is understanding the mechanisms of social variation across species because it can illuminate generalizable principles that might not be apparent in a single species. Most of what is known about the neural mechanisms of social variation comes from a single mammalian genus, *Microtus* rodents. Here, comparison of the pair-bonded to solitary phenotype has revealed the involvement of oxytocin, arginine vasopressin, dopamine and opioid neurochemical signalling. Differences in receptor density throughout specific regions of the brain are thought to gate social variation by tuning social recognition (nonapeptides), learned reward (dopamine) and positive affect (opioids) [2,4,7–9]. However, the extent to which neural patterns underlying social variation in voles translates to other taxa remains unclear.

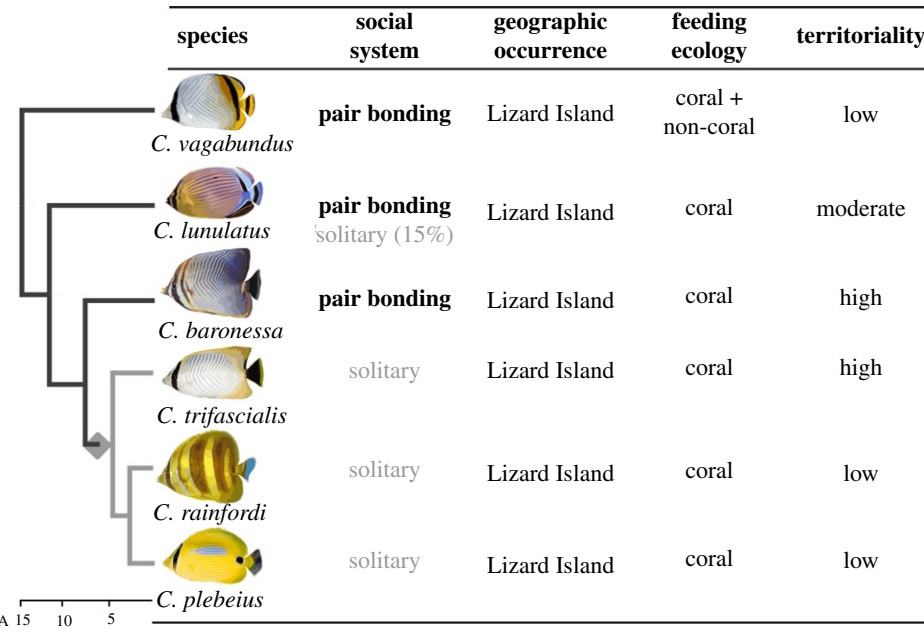

| species | social system | geographic occurrence | feeding ecology | territoriality |
|---|---|---|---|---|
| *C. vagabundus* | **pair bonding** | Lizard Island | coral + non-coral | low |
| *C. lunulatus* | **pair bonding** / solitary (15%) | Lizard Island | coral | moderate |
| *C. baronessa* | **pair bonding** | Lizard Island | coral | high |
| *C. trifascialis* | solitary | Lizard Island | coral | high |
| *C. rainfordi* | solitary | Lizard Island | coral | low |
| *C. plebeius* | solitary | Lizard Island | coral | low |

MYA 15    10    5

**Figure 1.** Study design comparing mechanisms of social variation in butterflyfishes. Within the populations, dichotomous social systems (pair-bonded versus solitary living) across individuals of *C. lunulatus* and across *Chaetodon* species do not covary with other attributes, offering more controlled examination of sociality. Grey diamond represents transition from pair-bonded to solitary living that occurred in the last common ancestor of *C. trifascialis*, *C. rainfordi* and *C. plebeius* at approximately 7 Ma. Table re-drawn from [3]. (Online version in colour.)

In vertebrates, the neurochemical–receptor systems and brain regions that regulate social behaviour are evolutionarily ancient and highly conserved [10,11]. It is therefore possible that similar patterns of neurochemical signalling within and among brain regions have been repeatedly recruited to facilitate social variation across different contexts. Nevertheless, sexual dimorphism in neural mechanisms of social variation may occur, due to sex differences in behavioural features of sociality and sex steroid hormones [12,13]. Sex differences may also arise from differences in pre-existing molecular and neural frameworks available for co-option during the development and/or evolution of sociality [13]. For example, in mammals, the evolutionary transition from solitude to pair bonding is hypothesized to have co-opted a pre-existing maternal–infant bond circuitry for females and territoriality circuitry for males [8]. These factors, in addition to phylogenetic distance, and the frequency and recency of evolutionary social transitions may also drive species differences in how the brain governs social variation [14]. Examining social variation in non-mammalian species is needed to better understand if its repeated evolution has relied on similar or labile neural mechanisms.

Coral reef butterflyfishes (f. Chaetodontidae) offer a compelling opportunity for comparatively studying the evolution of sociality [3,15,16]. Chaetodontidae have undergone rapid and repeated species diversification since the mid-Miocene [17], resulting in at least 127 extant species that co-occur in coral reefs throughout the world [18]. Unlike other vertebrate clades, pair bonding is ancestral and moderately conserved (77 of species; approx. 60%) [3], where it appears to have arisen due to the fitness benefits of cooperative territorial defence [19,20]. Pair bonds generally develop at the onset of reproductive maturity [21] and are characterized by a selective affiliation with one other individual, typically of the opposite sex [3]. Partners establish and defend stable coral feeding and refuge territories from neighbours using species-specific cooperative strategies [19,20]. Partnership

endurance is critical for this function [19], and has been reported to last for up to 7 years [22]. Yet, within the clade, several independent transitions to solitary living have occurred relatively recently [3]. Relative to pair bonders, 'singletons' display low levels and indiscriminate affiliation with other conspecifics, although the stability of solitary living throughout these individuals' lifetime remains unresolved [3]. Moreover, many species display variation in sociality across sympatric individuals, providing opportunities for intra-specific comparisons. In a previous study, we used these contrasts to develop a framework for studying within and between species variation in sociality [3] (figure 1). Specifically, in a wild population located on Lizard Island, Australia, we verified social variation (i.e. pair-bonded versus solitary living) both on an individual level within *Chaetodon lunulatu*s and across six closely related species (approx. 15 MYA divergence time) that represent a single evolutionary transition from pair-bonded (*C. vagabundus*, *C. lunulatus* and *C. baronessa*) to solitary (*C. trifascialis*, *C. rainfordii* and *C. plebeius*) living. Importantly, these differences in sociality do not covary with other major ecological attributes controlling for these potential confounds (figure 1). Nor is it likely to be confounded with parental care, which is a common confound in mammalian male pair-bonding systems that, although has not been studied in the focal populations, is reportedly absent in all butterflyfishes studied to date [20,23,24].

Here, we leveraged this butterflyfish system (figure 1) to compare brain gene expression associated with social variation (pair-bonded versus solitary living) between sexes and within versus among species. We focused on oxytocin- (*OTR*), vasopressin- (*V1aR*), dopaminergic- (*D1R*, *D2R*) and mu-opioid- (*MOR*) receptor gene expression within eight brain regions of the vertebrate social decision-making network [10], facilitating comparisons to mammals. We tested the hypothesis that in butterflyfishes, mechanisms of social variation are similar across sexes, and within versus between

species, and that neural patterns observed in butterflyfish are similar to those documented in *Microtus* voles.

## 2. Methods

### (a) Field collections

Specimens were collected from wild populations at Lizard Island, in the northern Great Barrier Reef (GBR), Australia, in 2013. Collections were made in May–July, outside of peak reproductive periods, capturing social behaviour independent of reproductive activity. Since we were interested in adults, only individuals that were within 80% of the asymptotic size for each species were used, as these are more likely to be reproductively mature [21]. Individuals were haphazardly encountered on snorkel, allowed to acclimate to observer presence for 3 min, and level of selective affiliation with another conspecific was observed for 3 min. Individuals displaying a high level of affiliation selectively with another conspecific were considered pair-bonded, while those displaying a low and indiscriminate level of affiliation with other conspecific were considered solitary. Collected sample sizes are as follows: *C. lunulatus* males = 11, females = 10; *C. baronessa* males = 7, females = 6; *C. vagabundus* males = 6, females = 6; *C. plebeius* females = 10; *C. rainfordii* males = 2, females = 11; *C. trifascialis* males = 2, females = 12. Behavioural characterization, collection and sexing of focal fishes have been described previously (see [3]). Brains were dissected, embedded in optimal cutting temperature compound (VWR), frozen and transported in liquid nitrogen, and stored at −80°C.

### (b) Gene expression

Frozen brains were coronally sectioned at 100 μm on a cryostat, and brain regions were identified using a *Chaetodon* brain atlas [25,26] and isolated using micropunches (figure 2 inset). RNA from each brain region was extracted and reverse transcribed to cDNA, purified and stored at −20°C. The *Chaetodon lunulatus* transcriptome was sequenced, from which cloning and exon-exon flanking qPCR primers for target genes and 18S endogenous control were designed and optimized (see electronic supplementary material, Materials). Quantitative PCR was then performed on each sample using a reaction mixture and qPCR cycling instrument (CFX380) that was recommended by the enzyme manufacturer (see electronic supplementary material, methods S1 for detailed methods, table S1 for primer sequences and table S2 for cycling parameters). Not all regions of each brain were measured for gene expression due to insufficient tissue available.

### (c) Statistical analysis

Gene expression differences were analysed for the *C. lunulatus* (intra-) and the *Chaetodon* (inter-) species comparison separately. Each brain region was analysed separately using the MCMC.qpcr model (MCMC.qpcr R package, v. 1.2.3 [27]), specifying the following parameter levels to optimize chain mixing: Cq1 = 49, number of iterations = 510 000, thinning interval = 500 and burnin = 10 000 (see electronic supplementary material, methods S2 for R script). For each analysis, gene expression differences were analysed by examining the main and interactive effects of sex and social system, with individual as a random factor. Results are reported as natural log fold changes of the posterior mean from an *a priori* reference state of male pair bonding. The *Chaetodon* species comparison revealed that for males, oxytocin receptor expression within the supracommissural part of the ventral telencephalon (Vs) varied significantly between social systems (see Results). In order to determine whether this difference was consistent across all species, we then examined the main effect of species exclusively for this brain region, gene

and sex. Results are reported as natural log fold changes in posterior mean from the *a priori* reference state of *C. baronessa*. Male *C. plebeius* were omitted from this analysis, as none were sampled. To approximate *p*-values based on a limited sample size, in all analyses, Bayesian two-sided *p*-values were calculated [27]. Pair-wise differences between factor levels were explored using the Tukey *post hoc* analysis.

## 3. Results

### (a) Pair-bonded versus solitary *Chaetodon lunulatus*

In *C. lunulatus*, brain region-specific receptor expression differed between social systems in sex-specific ways (electronic supplementary material, table S3; figure 2*a*). In *C. lunulatus* males, *D1R* within the ventral portion of the dorsal telencephalon (Vd, putative homologue of the mammalian nucleus accumbens) (ln(fold change) = −5.25; $p = 0.01$) and the preoptic area (POA) (ln(fold change) = −4.74; $p = 0.02$) was lower in solitary individuals than pair-bonded counterparts. *MOR* expression within the medial part of the dorsal telencephalon (Dm) was lower in solitary individuals than pair-bonded counterparts (ln(fold change) = −4.17, $p = 0.02$). Finally, *V1aR* expression within the ventral and lateral parts of the ventral telencephalon (Vv/Vl) appeared lower in solitary compared to pair-bonded individuals (ln(fold change) = −4.39); however, this was to a statistically non-significant extent ($p = 0.059$). In females, by contrast, *D1R* expression within the Dm (ln(fold change) = −3.91; $p = 0.03$) and Vd (ln(fold change) = −5.96; $p < 0.001$) was lower in solitary than pair-bonded individuals. Finally, *D2R* expression within the Vd was lower in solitary than pair-bonded individuals (ln(fold change) = −3.96; $p = 0.04$), and while Vd *OTR* showed a similar trend (ln(fold change) = −114.79), this was to a statistically non-significant extent ($p = 0.07$). See electronic supplementary material, table S3 for detailed statistical results.

### (b) Pair-bonded versus solitary *Chaetodon* species

Receptor expression patterning of individual differences in social systems found in *C. lunulatus* was inconsistent with that of species differences in the social system (figure 2*a,b*; electronic supplementary material, tables S3 and S4). Across sexes, genes and brain regions; only *OTR* expression within the supracommissural part of the ventral telencephalon (Vs, putative homologue of the mammalian medial amygdala/bed nucleus of the stria terminalis) varied consistently across all pair-bonded and solitary species in males, where expression was lower in solitary than pair-bonded males (ln(fold change) = −90.59; $p = 0.02$) (figure 2*b*). We found no brain region gene expression differences for *MOR* or dopamine receptors between pair-bonded and solitary species in either sex. See electronic supplementary material, tables S4 and S5 for detailed statistical results.

## 4. Discussion

Our work suggests that in *Chaetodon* butterflyfishes, nonapeptide, dopaminergic and mu-opioid receptor gene expression is important for mediating social variation. However, the brain regions in which signalling occurs differ by sex and within versus between species, mirroring previous findings in *Microtus* voles [4,7,28,29]. These results contribute to the

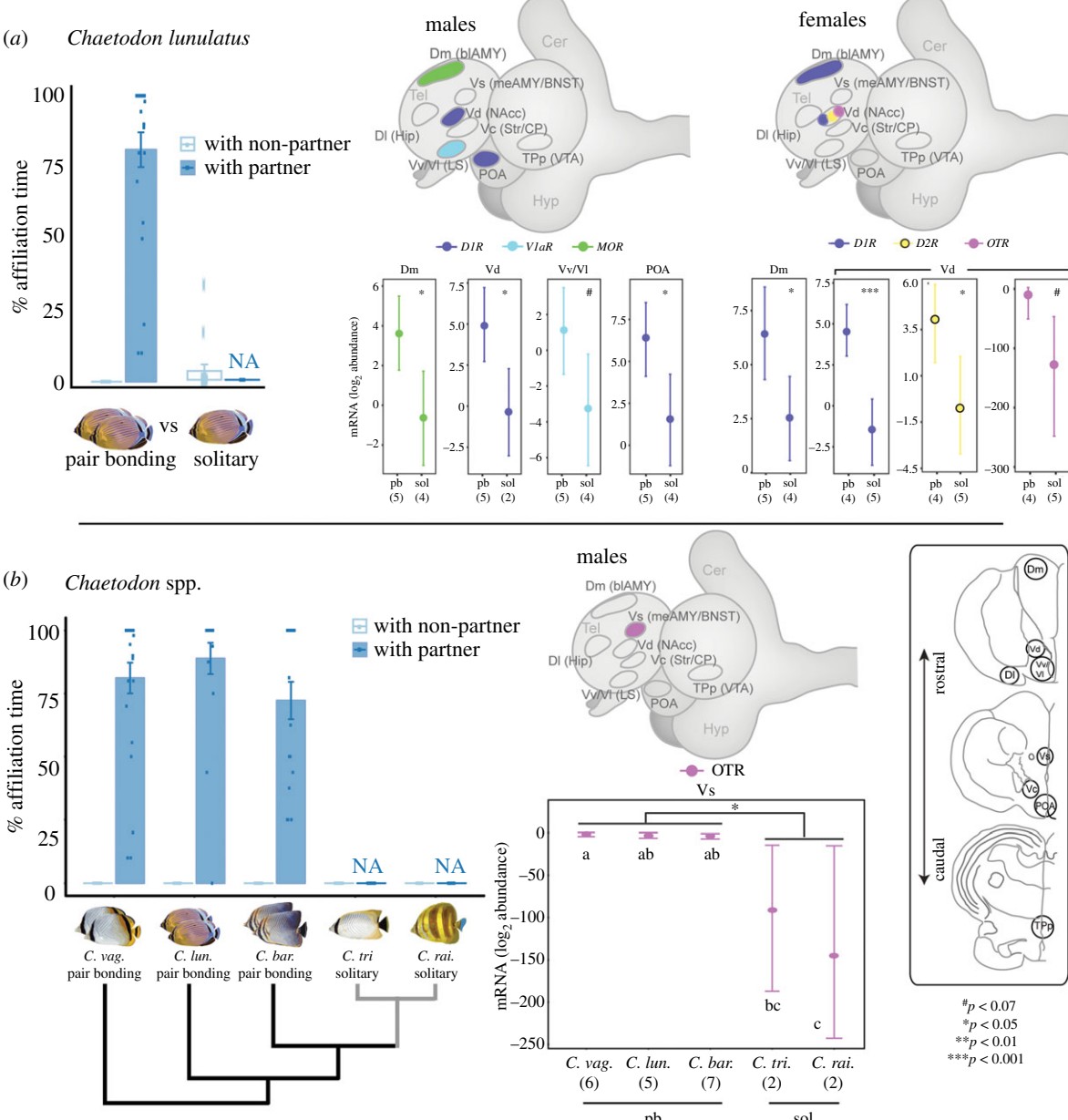

**Figure 2.** Neurochemical–receptor gene expression in specific brain regions differs between pair-bonded (pb) and solitary (sol) butterflyfishes in a context-specific manner. Gene expression of nonapeptide, dopamine and mu-opioid receptors correlates with social variation, yet patterns are distinct between sexes and (a) within versus (b) between species. Bar plots show pair-bonding fish displaying high levels of selective affiliation with a partner (blue) versus non-partner (white), and solitary fish displaying low levels, throughout a 3 min observation period immediately before collection. Behavioural values represent means ± s.e., and dots represent individual data points. Point plots show receptor gene expression within brain regions that differ significantly between social systems, where values represent means ± 95% credible intervals of the posterior distribution. Schematics show brain regions where receptor gene expression (light blue, arginine vasopressin *V1aR*; purple, oxytocin *OTR*; dark blue, dopamine *D1R*; yellow, dopamine *D2R*; green, mu-opioid *MOR*) differs between social systems. *Inset*: brain regions micro-dissected for gene expression analysis. *Brain region abbreviations:* teleost: telen., telencephalon; Dm, medial part of the dorsal telen.; Vd, dorsal part of the ventral telen.; Dl, lateral part of the dorsal telen.; Vv/Vl, lateral and ventral part of the ventral telen.; Vs, supracommissural part of the ventral telen.; Vc, central part of the ventral telen.; POA, pre optic area; TPp, periventricular part of the posterior tuberculum. Putative mammalian homologue: blAMY, basolateral amygdala; NAcc, nucleus accumbens; HIP, hippocampus; LS, lateral septum; meAMY/BNST, medial amygdala/bed nucleus of the stria terminalis; Str, Striatum; CP, caudate putamen; VTA, ventral tegmental area. (Online version in colour.)

emerging theme that the convergence of social variation across sexes and species has relied on repeated co-option of evolutionarily ancient nonapeptide, dopaminergic and opioid systems, though the precise signalling mechanisms can be evolutionarily labile.

Previous studies examining the involvement of receptor abundance patterns throughout the brain in individual social variation are largely limited to nonapeptide receptors in *Microtus ochrogaster* prairie voles, where patterns appear sexually dimorphic [4,7,28]. We found a similar trend in the oval

butterflyfish, where individual differences in sociality appear linked to differential *V1aR* expression in the ventral portion of the ventral telencephalon (Vv/vl) in males, and differential *OTR* expression in the dorsal portion of the ventral telencephalon (Vd) in females. These sex-specific patterns of individual social variation are distinct from voles and instead more closely mirror *species* differences, where monogamous species exhibit higher V1aR density within the lateral septum (LS, the putative mammalian homologue of the teleost Vv/Vl) in males [30], and higher OTR density within the nucleus accumbens

(NAcc, the putative mammalian homologue of the teleost Vd) in females and males [31]. We also examined the link between dopamine *D1R* and *D2R*, and *MOR* receptor expression and individual social variation, discovering that all three systems also appear involved in a sexually dimorphic manner. In females, forebrain *D1R* and *D2R* expression is involved; whereas in males, forebrain *D1R* and *MOR* expression is involved. Differential dopamine and MOR patterning within the brain is also linked to species variation in sociality in *Microtus* voles [31–33]. When compared with our current findings in more detail, two themes emerge: first, that NAcc-like D1R signalling is repeatedly involved; and second, the brain regions involved in MOR patterning are distinct. Taken together, these results suggest that lateral septum-like V1aR, nucleus accumbens-like OTR and D1R, and MOR signalling might have converged between rodents and fishes to mediate social variation, and this convergence is not specific to sex or level of social variation (i.e. inter-individual versus inter-species). Future studies should now examine whether these systems have also converged with rodents functionally, testing the hypothesis that social variation emerges from differential nonapeptide tuning of social recognition, mu-opioid tuning of positive hedonics and dopaminergic tuning the learned association between the two.

Can similar mechanistic modifications more *generally* explain repeated instances of social variation within and between species? In butterflyfishes, we found that neural patterning of social variation in oval butterflyfish did not translate across *Chaetodon* species, consistent with reports of prairie voles and their *Microtus* congeners (male *Microtus* NAcc OTR notwithstanding [4,7,28,29]). Consistently, certain neuromodulatory patterns of social variation found in a few rodent species do not translate more broadly across their genus [9,34,35]. Recent studies have compared multiple independent evolutionary social transitions across several species, in Lake Tanganyikan cichlids [36] and recently across vertebrates [37], and suggest that on a whole-brain level, repeated evolutionary transitions from promiscuous to monogamous living were accompanied by convergent changes in gene expression patterns. On a brain region-specific level, neuromodulation of social evolution across several species can be robust, as exemplified in estrildid finches [38] and butterflyfishes [15]. Yet, it can also be labile, even across relatively short divergence times and few transitional events. For example, two independent transitions from promiscuity to monogamy were accompanied by non-concordant V1aR patterning throughout the brain in *Peromyscus* mice [35]. Similarly, a single transition from monogamy to promiscuity was accompanied by non-concordant MOR patterning in *Microtus* voles [9]. In our study, we found that the transition from pair bonding to solitary living in six species of *Chaetodon* butterflyfishes was associated with non-concordant *MOR*, *D1R*, *D2R* and *V1aR* receptor expression patterning. Importantly, however, these negative results should be considered tentative, since the small sample size among solitary males likely resulted in limited experimental power. The only neuromodulatory pattern we found to be linked to species divergence in sociality is *OTR* expression in the Vs (teleost homologue of the mammalian medial amygdala/bed nucleus of the stria terminalis, meAMY/BNST), which is higher in pair-bonding species than solitary counterparts, specifically in males. This result mirrors that of *Microtus* voles (although in *Microtus* voles, this exists for both sexes) [31]. In male

rodents, amygdala activity is necessary for pair-bond affiliation [39], and OT signalling within the meAMY is critical for social recognition [40,41]. Whether amygdalar OTR signalling functions to gate species differences in sociality via tuning social recognition is an interesting line of inquiry for both butterflyfishes and rodents alike. Taken together, these studies suggest that neuromodulatory patterning of social evolution can be labile, highlighting the importance of comparing several species that ideally represent multiple independent social transitions when seeking generalizable mechanistic principles.

The differences in gene expression associated with social variation between sexes and within versus among species found in this study might be attributed to several factors. One may be differences in pre-existing neural phenotypes (e.g. level and location of ligand synthesis, projection pathways, receptor patterning) available for recruitment during the development and or evolution of sociality. For example, sexual dimorphism in nonapeptide expression across brain nuclei has been reported in teleosts [42,43], and we found that the trend for female-specific Vd *OTR* modulation of individual social variation coincides with higher baseline expression of this receptor within this brain region relative to males (electronic supplementary material, table S3). An additional potential source of variation is incomplete parallelism in behavioural details of social variation. For example, in *C. lunulatus*, while female pair bonders display lower territory defence than solitary counterparts; this is not observed in males [19]. Likewise, while some solitary individuals of *C. lunulatus* might represent 'divorcees' or 'widows' that were previously pair-bonded, this is highly unlikely for solitary species. Regardless of these inconsistencies, butterflyfishes represent an excellent research clade for understanding the evolution of neural mechanisms underlying sociality.

## 5. Conclusion

Social variation (i.e. pair-bonded versus solitary living) has repeatedly and independently evolved across vertebrates, the neural basis of which has been mostly resolved in mammals. We present one of the most comprehensive single studies on the neural basis of social variation outside of mammals, spanning several levels of mechanistic (i.e. from brain regions to receptor gene expression) and organismic (i.e. from individuals to species, in both sexes) organization in a phylogenetically distant lineage, teleost fishes. Our results suggest that nonapeptide, dopamine and mu-opioid receptor signalling within the forebrain mediates social variation among *Chaetodon* butterflyfishes, although the brain region-specific sites of action are context-specific. Taken together with previous studies in mammals, our results support the emerging idea that ancient nonapeptide, dopamine and mu-opioid systems have been repeatedly recruited to modulate social variation across both short and vast evolutionary distances in both sexes, albeit using different 'solutions' across different contexts [14]. A comprehensive assessment of neuromodulatory patterning throughout the brain across many taxonomically diverse species is now needed to further resolve major mechanistic themes of animal social variation.

Ethics. Animal collections followed Great Barrier Reef Marine Park Authority permit approvals: G10/33239.1, G13/35909.1, G14/37213.1; and James Cook University General Fisheries permit 170251. Animal handling and sacrifice procedures were approved by James Cook University Animal Ethics committee (approval: A1874).

Data accessibility. All data available from the Dryad Digital Repository: https://doi.org/10.5061/dryad.zcrjdfn7h [44].

Authors' contributions. J.P.N. conceived of the study; J.P.N., L.A.O., M.S.P. and S.P.W.W. designed and coordinated the study; J.P.N. and D.J.C. collected samples; J.P.N. and L.A.O. performed molecular work, J.P.N. analysed data; J.P.N. wrote the manuscript with input and editing from all authors. All authors gave final approval for publication and agree to be held accountable for the work performed therein.

Competing interests. We declare we have no competing interests.

Funding. This work was supported by the ARC Centre of Excellence for Coral Reef Studies (grant no. CEO561435) through funds awarded to

M.S.P. and S.P.W.W., an NSF EDEN Research Travel Award to J.P.N. (grant no. 0955517) and a Harvard Bauer Fellowship and L'Oréal for Women in Science Award won by L.A.O. L.A.O. is a New York Stem Cell Foundation—Robertson Investigator.

Acknowledgements. We thank Manuela Giammusso for field assistance; Makhail Matz, Eva Fischer and Rayna Harris for statistical guidance; Adam Dewan for generously sharing pre-published drafts of a *Chaetodon* brain atlas; and the O'Connell lab for comments on earlier versions of the manuscript. We acknowledge the fishes that were used to conduct the present study. We thank two anonymous reviewers for feedback that improved the manuscript during the review process.

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
