## [Reviewer comments · Proceedings of the Royal Society B: Biological Sciences]

Review History

RSPB-2020-0239.R0 (Original submission)

Review form: Reviewer 1

Recommendation

Accept with minor revision (please list in comments)

Scientific importance: Is the manuscript an original and important contribution to its field?

Good

General interest: Is the paper of sufficient general interest?

Good

Quality of the paper: Is the overall quality of the paper suitable?

Good

Is the length of the paper justified?

Yes

Should the paper be seen by a specialist statistical reviewer?

No

Do you have any concerns about statistical analyses in this paper? If so, please specify them explicitly in your report.

No

It is a condition of publication that authors make their supporting data, code and materials available - either as supplementary material or hosted in an external repository. Please rate, if applicable, the supporting data on the following criteria.

Is it accessible?

Yes

Is it clear?

Yes

Is it adequate?

Yes

Do you have any ethical concerns with this paper?

No

Comments to the Author

This paper makes use of quantitative PCR to measure mRNA expression of neuromodulatory receptors in the brain of several species of butterflyfish, focusing on regions associated with social decision making. The authors find intraspecific variation in nonapeptide receptor expression related to social phenotype within one species, though receptor types and brain regions that differed in expression varied by sex. They also show that expression of one receptor, OTR, in a single brain region, Vs, consistently varies by social phenotype across several Chaetodon species. This work builds upon studies of neural correlates of sociality in other taxa, particularly *Microtus* voles, and expands it to a group of teleost fish. That the authors find the same types of receptors vary across species and individuals within species expressing different social phenotypes, but that the pattern of expression is highly species-specific and does not necessarily predict species-level patterns of sociality mirrors results from work in other taxa and is informative of an evolutionary trend. This work should be of broad interest to the fields of animal social behavior and social neuroscience. However, I have several concerns, listed below, that should be addressed before publication.

In general, I am concerned with the small sample size of males among solitary species. Increasing the sample size for these comparisons would strengthen the paper significantly. In the absence of increased sampling, any negative result must be carefully interpreted and clearly account for lack of statistical power.

Title-“Neural correlates” typically refers to circuitry and/or cell activity related to behavior. As neither are investigated here, the title should be revised.

L78-typo “one onother”

L78-typo “opposit sex”

Figure 1-Because parental care is not studied in focal populations and only inferred, the Parental Care column should be removed from the table

L93-If parental care/lack of care is unstudied in these species, the authors cannot claim that paternal care does not vary with sociality

L135-typo "100 μ mon"

L145-161-Sample sizes for *C. rainfordii* and *C. trifascialis* are very low (n=2 each). Statistical analysis section should describe how small sample size was accounted for in analyses.

L165-typo "social systemsin"

Figure 2-Total observation period of collected individuals used to determine affiliation time should be reported in Field Collections section of Methods and Figure 2 caption

Figure 2-Sample size for mRNA abundance comparisons need to be shown in the figure and preferably individual data points displayed

L238-264-Should also acknowledge work investigating role of nonapeptide systems in interspecies differences in sociality (territoriality vs. gregariousness) in estrildid finches

L260-should read "species differences in sociality"

Review form: Reviewer 2

Recommendation

Accept with minor revision (please list in comments)

Scientific importance: Is the manuscript an original and important contribution to its field?

Excellent

General interest: Is the paper of sufficient general interest?

Excellent

Quality of the paper: Is the overall quality of the paper suitable?

Excellent

Is the length of the paper justified?

Yes

Should the paper be seen by a specialist statistical reviewer?

No

Do you have any concerns about statistical analyses in this paper? If so, please specify them explicitly in your report.

No

It is a condition of publication that authors make their supporting data, code and materials available - either as supplementary material or hosted in an external repository. Please rate, if applicable, the supporting data on the following criteria.

Is it accessible?

Yes

Is it clear?

Yes

Is it adequate?

Yes

Do you have any ethical concerns with this paper?

No

Comments to the Author

The authors aim to study the neural correlates of variation in sociality in a vertebrate. To do so, they use an interesting comparative approach using an ideal system: the butterfly fishes. These fish exhibit variation in their sociality both within and between species. The authors wanted to test if nonapeptides implicated in affiliative behaviours in mammals were also variable in other vertebrates (fishes) in individuals that differed in their sociality. Studying a fish species also have the advantage of telling us more about this group of vertebrates, which (let's not forget) is composed of more than 25 000 species and needs to be studied in its own right!

The authors used two different comparative approaches: they studied individuals that were pair bonding and compared them with individuals that did not, and also compared species that showed high affiliation with species that did not. This allowed them to see if the neural substrate of this behaviour is specific to each transition and comparable when looking at a species divergence versus within species individual variation. Furthermore, they studied both sexes, to test if males and females show unique association between receptor spatial distribution in the brain and affiliative behavior, as is found in mammals.

I found this article very clearly written. Each sentence had a purpose, within a very well defined structure. The manuscript is written for a general audience, while also providing all the information necessary for experts in the field.

The dataset and model are very novel and open a new window into our understanding of sociality. The results obtained are exciting and there is a wealth of follow up questions that will result from this work.

I was impressed with the data visualization, which transformed potentially complicated results (there are a lot of interactions between factors) into a clear portrait of the patterns observed with these two complimentary approaches. In association, I found the statistical analyses very solid and easy to follow (easy as can be for a quite complex set-up).

The discussion highlighting which patterns are specific and which ones are similar to mammals help us understand the molecular path that evolution takes during the course of the evolution of social behaviour.

Overall, this is one of the highest quality papers I have read in recent years.

I have very minor comments

P.2

Line 41 (abstract)

Would it be better for the reader to state in which directions these associations go?

p.5

line 135

put a space after "100 um"

line 165

add a space after "social systems"

Decision letter (RSPB-2020-0239.R0)

19-Mar-2020

Dear Dr Nowicki:

Your manuscript has now been peer reviewed and the reviews have been assessed by an Associate Editor. The reviewers' comments (not including confidential comments to the Editor) and the comments from the Associate Editor are included at the end of this email for your reference. As you will see, the reviewers and the Editors have raised some concerns with your manuscript and we would like to invite you to revise your manuscript to address them.

Importantly, please add mention, as per a reviewer's comment, of the limitation that there is "a low sample size for males of species representing the solitary clade".

Research ethics:

Use of animals and field studies:

It is a condition of publication that you make available the data and research materials supporting the results in the article. Datasets should be deposited in an appropriate publicly available repository and details of the associated accession number, link or DOI to the datasets

must be included in the Data Accessibility section of the article (<https://royalsociety.org/journals/ethics-policies/data-sharing-mining/>). Reference(s) to dataset(s) should also be included in the reference list of the article with DOIs (where available).

Please submit a copy of your revised paper within three weeks. If we do not hear from you within this time your manuscript will be rejected. If you are unable to meet this deadline please let us know as soon as possible, as we may be able to grant a short extension.

Best wishes,
Dr John Hutchinson, Editor
<mailto:proceedingsb@royalsociety.org>

Associate Editor
Board Member: 1
Comments to Author:

We have received two reviews for your manuscript "Neural correlates of social variation in coral reef butterflyfishes" which were over-all quite positive. However, one major statistical concern and other minor concerns were provided. Therefore, we are providing you the opportunity to review their comments and revise the manuscript accordingly.

Reviewer(s)' Comments to Author:

Referee: 1

Comments to the Author(s)

This paper makes use of quantitative PCR to measure mRNA expression of neuromodulatory receptors in the brain of several species of butterflyfish, focusing on regions associated with social decision making. The authors find intraspecific variation in nonapeptide receptor expression related to social phenotype within one species, though receptor types and brain regions that differed in expression varied by sex. They also show that expression of one receptor, OTR, in a single brain region, Vs, consistently varies by social phenotype across several *Chaetodon* species. This work builds upon studies of neural correlates of sociality in other taxa, particularly *Microtus* voles, and expands it to a group of teleost fish. That the authors find the same types of receptors vary across species and individuals within species expressing different social phenotypes, but that the pattern of expression is highly species-specific and does not necessarily predict species-level patterns of sociality mirrors results from work in other taxa and is informative of an evolutionary trend. This work should be of broad interest to the fields of animal social behavior and social neuroscience. However, I have several concerns, listed below, that should be addressed before publication.

In general, I am concerned with the small sample size of males among solitary species. Increasing the sample size for these comparisons would strengthen the paper significantly. In the absence of increased sampling, any negative result must be carefully interpreted and clearly account for lack of statistical power.

Title—"Neural correlates" typically refers to circuitry and/or cell activity related to behavior. As neither are investigated here, the title should be revised.

L78-typo "one onother"

L78-typo "opposit sex"

Figure 1-Because parental care is not studied in focal populations and only inferred, the Parental Care column should be removed from the table

L93-If parental care/lack of care is unstudied in these species, the authors cannot claim that paternal care does not vary with sociality

L135-typo "100 μ mon"

L145-161-Sample sizes for *C. rainfordii* and *C. trifascialis* are very low (n=2 each). Statistical analysis section should describe how small sample size was accounted for in analyses.

L165-typo "social systemsin"

Figure 2-Total observation period of collected individuals used to determine affiliation time should be reported in Field Collections section of Methods and Figure 2 caption

Figure 2-Sample size for mRNA abundance comparisons need to be shown in the figure and preferably individual data points displayed

L238-264-Should also acknowledge work investigating role of nonapeptide systems in interspecies differences in sociality (territoriality vs. gregariousness) in estrildid finches

L260-should read "species differences in sociality"

Referee: 2

Comments to the Author(s)

The authors aim to study the neural correlates of variation in sociality in a vertebrate. To do so, they use an interesting comparative approach using an ideal system: the butterfly fishes. These fish exhibit variation in their sociality both within and between species. The authors wanted to test if nonapeptides implicated in affiliative behaviours in mammals were also variable in other vertebrates (fishes) in individuals that differed in their sociality. Studying a fish species also have the advantage of telling us more about this group of vertebrates, which (let's not forget) is composed of more than 25 000 species and needs to be studied in its own right!

The authors used two different comparative approaches: they studied individuals that were pair bonding and compared them with individuals that did not, and also compared species that showed high affiliation with species that did not. This allowed them to see if the neural substrate of this behaviour is specific to each transition and comparable when looking at a species divergence versus within species individual variation. Furthermore, they studied both sexes, to test if males and females show unique association between receptor spatial distribution in the brain and affiliative behavior, as is found in mammals.

I found this article very clearly written. Each sentence had a purpose, within a very well defined structure. The manuscript is written for a general audience, while also providing all the information necessary for experts in the field.

The dataset and model are very novel and open a new window into our understanding of sociality. The results obtained are exciting and there is a wealth of follow up questions that will result from this work.

I was impressed with the data visualization, which transformed potentially complicated results (there are a lot of interactions between factors) into a clear portrait of the patterns observed with these two complimentary approaches. In association, I found the statistical analyses very solid and easy to follow (easy as can be for a quite complex set-up).

The discussion highlighting which patterns are specific and which ones are similar to mammals help us understand the molecular path that evolution takes during the course of the evolution of social behaviour.

Overall, this is one of the highest quality papers I have read in recent years.

I have very minor comments

P.2

Line 41 (abstract)

Would it be better for the reader to state in which directions these associations go?

p.5

line 135

put a space after "100 um"

line 165

add a space after "social systems"

Author's Response to Decision Letter for (RSPB-2020-0239.R0)

See Appendix A.

Decision letter (RSPB-2020-0239.R1)

07-Apr-2020

Dear Dr Nowicki

I am pleased to inform you that your manuscript entitled "Gene expression correlates of social evolution in coral reef butterflyfishes" has been accepted for publication in Proceedings B. Congratulations!!

Open Access

You are invited to opt for Open Access, making your freely available to all as soon as it is ready for publication under a CCBY licence. Our article processing charge for Open Access is £1700. Corresponding authors from member institutions (<http://royalsocietypublishing.org/site/librarians/allmembers.xhtml>) receive a 25% discount to these charges. For more information please visit <http://royalsocietypublishing.org/open-access>.

Your article has been estimated as being 7 pages long. Our Production Office will be able to confirm the exact length at proof stage.

Paper charges

Sincerely,
Dr John Hutchinson
Editor, Proceedings B

Author's Response to Decision Letter for (RSPB-2020-0239.R1)

See Appendix B.

Decision letter (RSPB-2020-0239.R2)

01-Jun-2020

Dear Dr O'Connell

I am pleased to inform you that your manuscript entitled "Gene expression correlates of social evolution in coral reef butterflyfishes" has been accepted for publication in Proceedings B. Congratulations!!

Open Access

You are invited to opt for Open Access, making your freely available to all as soon as it is ready for publication under a CC BY licence. Our article processing charge for Open Access is £1700. Corresponding authors from member institutions (<http://royalsocietypublishing.org/site/librarians/allmembers.xhtml>) receive a 25% discount to these charges. For more information please visit <http://royalsocietypublishing.org/open-access>.

Your article has been estimated as being 7 pages long. Our Production Office will be able to confirm the exact length at proof stage.

Paper charges

Sincerely,

Dr John Hutchinson
Editor, Proceedings B
mailto: proceedingsb@royalsociety.org

Associate Editor:

Board Member

Comments to Author:

It was commendable of the authors to bring in an external statistician to revisit the assumption of their model. The manuscript is re-accepted.

Appendix A

AUTHOR RESPONSE TO EDITOR AND REVIEWERS (IN BLUE)

Line numbers (LXX) correspond to the main document that will be re-submitted with tracked changes on, entitled, "main_document_resubmit_1_track_changes".

Associate Editor

Board Member: 1

Comments to Author:

We have received two reviews for your manuscript "Neural correlates of social variation in coral reef butterflyfishes" which were over-all quite positive. However, one major statistical concern and other minor concerns were provided. Therefore, we are providing you the opportunity to review their comments and revise the manuscript accordingly.

Reviewer(s)' Comments to Author:

Referee: 1

Comments to the Author(s)

This paper makes use of quantitative PCR to measure mRNA expression of neuromodulatory receptors in the brain of several species of butterflyfish, focusing on regions associated with social decision making. The authors find intraspecific variation in nonapeptide receptor expression related to social phenotype within one species, though receptor types and brain regions that differed in expression varied by sex. They also show that expression of one receptor, OTR, in a single brain region, Vs, consistently varies by social phenotype across several Chaetodon species. This work builds upon studies of neural correlates of sociality in other taxa, particularly *Microtus* voles, and expands it to a group of teleost fish. That the authors find the same types of receptors vary across species and individuals within species expressing different social phenotypes, but that the pattern of expression is highly species-specific and does not necessarily predict species-level patterns of sociality mirrors results from work in other taxa and is informative of an evolutionary trend. This work should be of broad interest to the fields of animal social behavior and social neuroscience. However, I have several concerns, listed below, that should be addressed before publication.

In general, I am concerned with the small sample size of males among solitary species. Increasing the sample size for these comparisons would strengthen the paper significantly. In the absence of increased sampling, any negative result must be carefully interpreted and clearly account for lack of statistical power.

Thank you for this consideration. We agree. We had set out to collect N = 8 individuals per sex per experimental group. However, this proved challenging, due to an unexpected female-biased sex ratio among solitary species within the wild population, and the inability to sex butterflyfishes pre-mortem. By the time we realized this limitation, we were unable to return to the field site to increase sample size, due to restraints on funds and the collection permit. As per reviewer suggestion, we have therefore amended the text in the discussion (L272) to state, "Importantly, however, these negative results should be interpreted cautiously and considered tentative, since the small sample size among solitary males likely resulted in limited experimental power." Additionally, we kept the original text that reinforced this limitation in the conclusion (L310), "That neither dopaminergic nor opioid expression was linked to social variation in this study does not preclude the possibility of their involvement, which future studies might

capture with higher sample size, a more comprehensive survey of brain regions, or analysis of receptor protein rather than mRNA.”

Title-“Neural correlates” typically refers to circuitry and/or cell activity related to behavior. As neither are investigated here, the title should be revised.

We acknowledge this and have changed the title to “Gene expression correlates of social evolution in coral reef butterflyfishes”. (L1)

L78-typo “one onother”

Corrected to “one other”, thank you. (L84)

L78-typo “opposit sex”

Corrected to “opposite sex”, thank you. (L84)

Figure 1-Because parental care is not studied in focal populations and only inferred, the Parental Care column should be removed from the table

We have done so.

L93-If parental care/lack of care is unstudied in these species, the authors cannot claim that paternal care does not vary with sociality

We appreciate this concern and agree. We have amended the text to be more cautious: “Nor is it likely to be confounded with parental care--a common confound in mammalian pair bonding systems that although hasn't been studied in the focal population, is reportedly absent in all butterflyfishes studied to date [20, 23,24]. (L100)

L135-typo “100 μ mon”

Corrected to “100 μ m on”, thank you. (L147)

L145-161-Sample sizes for *C. rainfordii* and *C. trifascialis* are very low (n=2 each). Statistical analysis section should describe how small sample size was accounted for in analyses.

We have amended text to do so (L169): “To approximate p-values based on a limited sample size, in all analyses, Bayesian two-sided p-values were calculated using the posterior distribution tested with a z-score against a normal distribution [29]. Z-score based p-values were then adjusted for multiple hypothesis testing using the False Discovery Rate method [30].

L165-typo “social systemsin”

Corrected to “social systems in”, thank you. (L178)

Figure 2-Total observation period of collected individuals used to determine affiliation time should be reported in Field Collections section of Methods and Figure 2 caption.

We have done so, thank you.

Methods: “Individuals were haphazardly encountered on snorkel, allowed to acclimate to observer presence for 3 min, and level of selective affiliation with another conspecific was observed for 3 min.” (L131).

Figure 2 caption: “Bar plots show pair bonding fish displaying high levels of selective affiliation with a partner (blue) vs. non-partner (white), and solitary fish displaying low levels, throughout a 3 min observation period immediately before collection. (L461)

Figure 2-Sample size for mRNA abundance comparisons need to be shown in the figure and preferably individual data points displayed.

We have included the sample sizes in the figure, thank you. In doing so, we identified an error in the statistical analysis that led to a false positive in one result, specifically OTR within the Vd (NAcc) in *Chaetodon lunulatus* males. After correct re-analysis, we found no significant difference in OTR within the Vd (NAcc) in *C. lunulatus* males. We have corrected this error in the manuscript by amending all relevant areas (main and supplementary results, figure, discussion) accordingly. Please note that this correction does not change the major findings, conclusions, or narrative of the manuscript. Necessary text modifications were minor, and are as follows:

RESULTS:

Original text that was removed “...and OTR gene expression within the dorsal part of the ventral telencephalon (Vd, putative homolog of the mammalian nucleus accumbens) ($\ln(\text{fold change}) = -73.867$; $p_{\text{adj-z}} = 0.01$)”. (L181)

Accompanying Supplementary material results: updated.

FIGURE 2:

The original incorrect result has been removed from this figure.

DISCUSSION:

Correction 1 (in red) to original text (in blue):

Pair bonded males exhibit different Vc OTR expression relative to solitary counterparts, mirroring paired prairie vole males that exhibit higher nucleus accumbens (NAcc, a sub-structure of striatum, which is the putative mammalian homologue of the teleost Vc) OTR density relative to solitary ‘wanderers’ [7]. OT-OTR binding within the striatal NAcc is critical for male pair bond formation in prairie voles and may gate the learned association between partner identity and hedonic reward, effectively biasing males to either form pair bonds or remain single [7,35]. Whatever the function of striatal NAcc OTR signaling might be, these parallels across phylogenetically distant species tentatively suggest that striatal-like OTR expression represents a convergent mechanism of individual social variation in males. (L241-250)

L238-264-Should also acknowledge work investigating role of nonapeptide systems in interspecies differences in sociality (territoriality vs. gregariousness) in estrildid finches.

We thank the reviewer for this suggestion and agree. We have edited text to read, “On a brain region-specific level, neuromodulation of social evolution across several species can be robust, as exemplified in Estrildid finches [40] and butterflyfishes [15].” (L261)

L260-should read “species differences in sociality”

Done, thank you.

Referee: 2

Comments to the Author(s)

The authors aim to study the neural correlates of variation in sociality in a vertebrate. To do so, they use an interesting comparative approach using an ideal system: the butterfly fishes. These fish exhibit variation in their sociality both within and between species. The authors wanted to test if nonapeptides implicated in affiliative behaviours in mammals were also variable in other vertebrates (fishes) in individuals that differed in their sociality. Studying a fish species also have the advantage of telling us more about this group of vertebrates, which (let's not forget) is composed of more than 25 000 species and needs to be studied in its own right!

The authors used two different comparative approaches: they studied individuals that were pair bonding and compared them with individuals that did not, and also compared species that showed high affiliation with species that did not. This allowed them to see if the neural substrate of this behaviour is specific to each transition and comparable when looking at a species divergence versus within species individual variation. Furthermore, they studied both sexes, to test if males and females show unique association between receptor spatial distribution in the brain and affiliative behavior, as is found in mammals.

I found this article very clearly written. Each sentence had a purpose, within a very well defined structure. The manuscript is written for a general audience, while also providing all the information necessary for experts in the field.

The dataset and model are very novel and open a new window into our understanding of sociality. The results obtained are exciting and there is a wealth of follow up questions that will result from this work.

I was impressed with the data visualization, which transformed potentially complicated results (there are a lot of interactions between factors) into a clear portrait of the patterns observed with these two complimentary approaches. In association, I found the statistical analyses very solid and easy to follow (easy as can be for a quite complex set-up).

The discussion highlighting which patterns are specific and which ones are similar to mammals help us understand the molecular path that evolution takes during the course of the evolution of social behaviour.

Overall, this is one of the highest quality papers I have read in recent years.

We are very pleased that the reviewer thinks highly of our paper! Thank you!

I have very minor comments

P.2

Line 41 (abstract)

Would it be better for the reader to state in which directions these associations go?

Thank you for this consideration. We agree and have amended text to state, "Here, *OTR* expression within the supracommissural part of the ventral telencephalon was higher in pair bonded than solitary species, specifically for males." (L 41).

p.5

line 135

put a space after "100 um"

Done, thank you. (L147)

line 165

add a space after “social systems”

Done, thank you. (L179)

Journal Name: Proceedings of the Royal Society B Journal Code: RSPB Print ISSN: 0962-8452 Online ISSN: 1471-2954 Journal Admin Email: proceedingsb@royalsociety.org MS Reference Number: RSPB-2020-0239 Article Status: SUBMITTED MS Dryad ID: RSPB-2020-0239 MS Title: Neural correlates of social variation in coral reef butterflyfishes MS Authors: Nowicki, Jessica; Pratchett, Morgan; Coker, Darren; Walker, Stefan; O'Connell, Lauren Contact Author: Jessica Nowicki Contact Author Email: jnowicki@stanford.edu Contact Author Address 1:

Contact Author Address 2:

Contact Author Address 3:

Contact Author City: Stanford

Contact Author State: California

Contact Author Country: United States

Contact Author ZIP/Postal Code:

Keywords: pair bond, sex differences, species differences, butterflyfish, nonapeptides

Abstract: Animals display remarkable variation in social behavior. However, outside of rodents, little is known about the neural mechanisms of social variation, and whether they are shared within and across species and sexes, limiting our understanding of how sociality evolves. Using coral reef butterflyfishes, we examined the neural correlates of social variation (i.e., pair bonding vs. solitary living) within and between species and sexes. In several brain regions, we quantified gene expression of receptors important for social variation in mammals: oxytocin (OTR), arginine vasopressin (V1aR), dopamine (D1R, D2R), and mu-opioid (MOR). We found that social variation across individuals of the oval butterflyfish, *Chaetodon lunulatus*, is linked to differences in OTR and V1aR gene expression within several forebrain regions in a sexually dimorphic manner. However, this contrasted with social variation among six species representing a single evolutionary transition from pair bonded to solitary living. Here, OTR expression within the supracommissural part of the ventral telencephalon predicted inter-specific social variation, specifically in males. These results contribute to the emerging idea that nonapeptide signaling is a central theme to the evolution of sociality across individuals, although the precise mechanism may be flexible across sexes and species.

EndDryadContent

Appendix B

Dear Dr. Hutchinson and/or referees:

We are pleased to submit a revised version of our manuscript “Gene expression correlates of social evolution in coral reef butterflyfishes” by Jessica P. Nowicki, Morgan S. Pratchett, Stefan P. W. Walker, Darren J. Coker, and Lauren A. O’Connell as a new research article for Proceeding of the Royal Society of London B (PROC-B). We are pleased that the former manuscript was accepted for publication in PROC-B.

As you might remember, soon after acceptance, we discovered sub-optimized parameters in the statistical model (MCMCqpcr) that resulted in some results being unreproducible. We resolved this by working closely with the statistician who designed the model and wrote the associated R package. Specifically, we specified the following parameter levels to optimize chain mixing: Cq1 = 49, number of iterations = 510000, thinning interval = 500, and burnin = 10000; all of which have also been specified in the manuscript and supplementary material. We confirmed reproducibility by confirming that the optimized model yielded the same results when run 3 independent times. The statistician also confirmed that the multiple hypothesis testing correction used in the former manuscript was unnecessary, due to not enough comparisons being made to justify its use. We therefore corrected this error as well. We feel the manuscript has much improved. While some results are different than the former version, the overall conclusion and narrative is the same. Below is the revised article with ‘tracked changes’ on.

Thank you,
JPN